# Exquisitely Preserved Fossil Snakes of Messel: Insight into the Evolution, Biogeography, Habitat Preferences and Sensory Ecology of Early Boas

**Agustín Scanferla [1,2,\*]** and **Krister T. Smith [1,3]**

[1] Department of Messel Research and Mammalogy, Senckenberg Research Institute, Senckenberganlage 25, 60325 Frankfurt am Main, Germany; krister.smith@senckenberg.de

[2] CONICET-UNSa. Instituto de Bio y Geociencias del NOA (IBIGEO), 9 de Julio No. 14, A4405BBB Rosario de Lerma, Salta, Argentina

[3] Faculty of Biological Sciences, Institute for Ecology, Diversity and Evolution, Max-von-Laue-Strasse 13, University of Frankfurt, 60438 Frankfurt am Main, Germany

[*] Correspondence: agustin_scanferla@yahoo.com.ar

http://zoobank.org/urn:lsid:zoobank.org:act:8C99B8D8-243A-4CDB-B33D-EC37D4ED911D

**Abstract:** Our knowledge of early evolution of snakes is improving, but all that we can infer about the evolution of modern clades of snakes such as boas (Booidea) is still based on isolated bones. Here, we resolve the phylogenetic relationships of *Eoconstrictor fischeri* comb. nov. and other booids from the early-middle Eocene of Messel (Germany), the best-known fossil snake assemblage yet discovered. Our combined analyses demonstrate an affinity of *Eoconstrictor* with Neotropical boas, thus entailing a South America-to-Europe dispersal event. Other booid species from Messel are related to different New World clades, reinforcing the cosmopolitan nature of the Messel booid fauna. Our analyses indicate that *Eoconstrictor* was a terrestrial, medium- to large-bodied snake that bore labial pit organs in the upper jaw, the earliest evidence that the visual system in snakes incorporated the infrared spectrum. Evaluation of the known palaeobiology of *Eoconstrictor* provides no evidence that pit organs played a role in the predator–prey relations of this stem boid. At the same time, the morphological diversity of Messel booids reflects the occupation of several terrestrial macrohabitats, and even in the earliest booid community the relation between pit organs and body size is similar to that seen in booids today.

**Keywords:** Boidae; Messel Formation; Eocene; pit organs; infrared; macrohabitat; biogeography

## 1. Introduction

Snakes of the clade Boidae (boas, anacondas, emerald boas) are arguably among the most charismatic species of living reptiles. They are one of the first offshoots of that part of the snake tree that capture and ingest prey much larger than their own head through an arsenal of anatomical and behavioural features including constriction [1], macrostomy [2], and infrared detection as an integral part of their visual system [3,4]. Boid snakes, currently distributed in the Neotropics, are part of the larger clade Booidea (Neotropical boas, "erycines", Malagasy boas, ungaliophiines and Pacific island boas), which has fuelled much debate [5–9] as to how a reptile group of such low apparent vagility came to be distributed across all current continents except Antarctica [10]. Up to now, the fragmentary and questionable fossil record of booid snakes provides little insight into their early evolution and ecology.

The study of several exquisitely preserved skeletons of the booid snake *Eoconstrictor fischeri* from the Eocene Konservat-Lagerstätte of Messel (Germany) provides considerable new insight into the biology of early boas. In this paper we describe the anatomy of this species based on CT data sets

and analyse its phylogenetic relationships. We then discuss its implications for booid biogeography and the habitat preferences of this ancient boa. Finally, we present evidence for the early presence of specialised organs to detect infrared radiation and discuss its role in the ecological relations of this early boid relative.

## 2. Materials and Methods

### 2.1. Abbreviations

HLMD-Me, Messel collection, Hessisches Landesmuseum, Darmstadt, Germany.

SMF-ME, Messel vertebrate collection, Senckenberg Research Institute, Frankfurt am Main, Germany.

### 2.2. Computed Tomography (CT)

The skull of three specimens of *Eoconstrictor fischeri* (SMF-ME 2504a, SMF-ME 11332a, SMF-ME 11398) were micro-CT scanned. Specimen SMF-ME 11332a was scanned on a Tomoscope HV 500 (Werth Messtechnik GmbH) with a 2k detector and a 225-kV μ-focus X-ray source in the industrial μCT facility at the Technical University in Deggendorf, Germany. Scan parameters: CT mode 2, 150 μA, 190 kV, 1200 steps, voxel resolution 20.2 μm). Specimen SMF-ME 11398 was scanned on a aPhoenix v|tome|x scanner with a 1k detector at the Senckenberg Human Evolution and Palaeoenvironment CT laboratory at the University of Tübingen, Germany. Scan parameters: Multiscan mode (4 individual scans with 2500 steps each), Sector Scan mode over 278 °, 130 μA, 230 kV, voxel resolution 30.0 μm. Resulting volume files were analysed using VG Studio MAX v3.2 on a high-end workstation at Senckenberg.

### 2.3. Taxonomy

The new taxonomic name was registered with Zoobank and provided with an LSID .

### 2.4. Phylogenetic Analyses

We employed the morphological matrix of Smith and Scanferla [11]. We added several terminals that represent all living genera and well-known fossil Booidea. The resulting matrix with 201 osteological characters and 48 terminals was analysed in combination with DNA sequences for three mitochondrial (12S, 16S, Cytb) and five nuclear genes (BDNF, Cmos, NTF3, NGFB and PNN), all taken from the GenBank (accession codes available in electronic Supplementary Mterials; combined matrix is available as Data S1). We employed static homology via multiple alignment using default settings in Clustal X [12]. After alignment, each sequence was trimmed of its leading and lagging gaps. For maximum parsimony (MP) analyses we employed TNT [13]. All characters were equally weighted and treated as unordered, and gaps coded as missing data. Trees were rooted utilising the anguimorph lizard Varanus salvator as an outgroup. The search strategy employed in TNT was "Traditional search" (using TBR) with 1000 replications with the objective of encountering all possible tree islands. Two alternative support measures (Bremer support and bootstrap resampling) were calculated to evaluate the robustness of the nodes of the most parsimonious trees. Bootstrap values were calculated with 10,000 pseudoreplicates.

We furthermore conducted Bayesian inference (BI) using the fossilised birth-death process [14] as implemented in Mr. Bayes 3.2.1 [15]. For fossil taxa, a uniform prior between an upper and lower bound corresponding to the age uncertainty was applied. The analysis was performed with four chains in two independent runs with 40 million generations and tree sampling at every 1000 generations. A 25% burn-in rate was applied. To estimate divergence times, we applied the a posteriori time-scaling method of Bapst [16,17] using the package "paleotree" for R [18].

## 2.5. Infrared Organs Survey

The area of foramina usually reflects the amount of tissue that passes through them, e.g., [19]. Accordingly, the size of the foramina in the jaws is related to the number of nerve fibres and size of blood vessels that serve the sensory tissue of heat-sensing circumoral epithelium in snakes. In the upper jaw this tissue is innervated by the maxillary and/or ophthalmic branch of the trigeminal, and perfused by branches of the superior maxillary artery, which pass through the maxillary labial foramina in squamates [20,21]. In the lower jaw, this tissue is innervated by the mandibular branch of the trigeminal, and perfused by branches of the mandibular artery, which pass through the mental and anterior surangular foramina in snakes [20].

To determine whether pit organs were present in the Messel fossil booids, including *Eoconstrictor fischeri*, we gathered information on the presence of pits for 27 extant species in five regions of the jaws: rostral, anterior supralabial, posterior supralabial, anterior infralabial, and posterior infralabial. We then measured the dorsoventral height of the foramina in the jaws (electronic Supplementary Mterials, Table S1). As suggested by Kluge [22], the height of foramina will not be affected by the orientation of the foramina in the horizontal plane. We focused on boas and pythons because, in contrast to Viperidae, they retain a more plesiomorphic maxillary morphology. We also measured two colubroids (*Lampropeltis getulus* and *Thamnophis marcianus*), which were surveyed by Barrett et al. [23] with regard to their sensitivity to radiant heat. We estimated the cross-sectional area of each foramen from its height and then summed those area values for all foramina in the maxilla and for both foramina in the dentary in order to estimate the total amount of neurovascular tissue related to the circumoral area of a jaw quadrant. While it should theoretically be possible to study the relation between foramen size and the presence of pit organs only in that part of the jaw innervated by the nerve, uncertainty and known variation in the innervation pattern led us to treat each jaw quadrate as a whole using the summed area. From the summed area, we then calculated the (1-dimensional) theoretical diameter of a single foramen for the upper and for the lower jaw that would have contained this tissue. Finally, we normalised this theoretical diameter by dividing it by the skull length, as measured from snout tip to the posterior end of the occipital condyle. We used a kind of generalised linear model, logistic regression, to model the correlation between theoretical, normalised foramen diameter and the presence of pits. Based on the model, we then calculated the probability that pits were present in the upper and lower jaws of the Messel fossil snakes, given the theoretical, normalised diameter of the foramina in I (SMF-ME 11398), *Messelophis variatus* (SMF-ME 1828), *Rieppelophis ermannorum* (HLMD-Me 7915), and *Rageryx schmidi* (HLMD-Me 9723).

## 2.6. Habitat Preference Survey

In extant snakes, there is a correlation between body size and tail length, on the one hand, and habitat preferences [24]. Sheehy et al. [24] assigned all species in their data-set to one of four categories: aquatic, nonscansorial ("ground-dwelling" here), eurytopically arboreal/terrestrial ("generalist" here), and stenotopically arboreal ("arboreal" here). To explore the significance of these conclusions for the Messel fossil taxa, we took the dataset of Sheehy et al. [24], added measurements on 9 species (for a total of 234 species; see electronic Supplementary Mterials, Table S2), and used it to calculate snout-vent length (SVL) and absolute tail length. We also took estimates of these variables from Schaal [25] for *Eoconstrictor fischeri* (SMF-ME 2504), from Schaal and Baszio [26] for *Rieppelophis ermannorum* (HLMD-Me 7915), from Baszio [27] for *Messelophis variatus* (HLMD-Me 15013), and from Smith and Scanferla [11] for *Rageryx schmidi* (HLMD-Me 9723). We calculated the common logarithm of all measurements and conducted principle components analysis (PCA) on the logged data. We also conducted phylogenetic PCA (pPCA) on the logged data based the tree of Pyron et al. [28], pruned to the species sampling in our data set. We followed a Brownian model of evolution, as our estimates of Pagel's lambda were not very far from 1 for either SVL or tail length. It was not possible to include the fossil taxa in the pPCA, because they are not present in the Pyron et al. [28] tree. However, the results of pPCA were highly similar to those of the PCA, so that individual extant species are in many cases

readily identifiable in the two plots of PC1 and 2, so we do not believe that pPCA would produce different results with respect to the fossils. For pPCA we made use of the function "phyl.pca" in the phytools package for R [29]. We also conducted linear discriminant analysis (LDA) on the logged data of extant species, as these were more closely normally distributed. We then used the LDA model to predict, with default assumptions, the habitat preferences of the Messel fossil taxa. All calculations were performed in R v.3.5 [18].

## 3. Results

### 3.1. Systematic Palaeontology

#### 3.1.1. Genus-level taxonomy

Serpentes Linnaeus, 1758
Alethinophidia Nopcsa, 1923
Booidea Gray, 1825 sensu Pyron, Reynolds and Burbrink, 2014 [30]
Genus *Eoconstrictor* Scanferla and Smith nov.
http://zoobank.org/urn:lsid:zoobank.org:act:2DF6D730-E3C7-4F96-89BA-6BF91E708562
**Type and only known species**. *Eoconstrictor fischeri* (Schaal, 2004) comb. nov.
**Etymology**. *Eo* ('Εως in Greek): in Greek mythology, the goddess that brought the dawn to Earth every morning; constrictor (Latin): one who constricts.
**Diagnosis**. As for type and only known species.

**Remarks**: *Currently all larger "booid" vertebrae from the early Palaeogene of Europe have been referred to species of the genera Palaeopython or Paleryx [31]. However, ongoing revision of these genera based on the type material, including cranial elements, shows that "Palaeopython" fischeri is not closely related to the type species of Palaeopython, P. cadurcensis, and lacks diagnostic features of the type species of Paleryx, P. rhombifer. Thus, we consider that fischeri represents a distinct lineage and requires a new generic name.*

#### 3.1.2. Species-level taxonomy

*Eoconstrictor fischeri* (Schaal, 2004) comb. nov.
**Holotype**. SMF-ME 929, seven mid-trunk vertebrae.
**Referred specimens**. Specimen with crocodylian in gut (accessioned in the Fossilien- und Heimatmuseum Messel; no number); SMF-ME 1002, skeleton without skull; SMF-ME 2504, skeleton; SMF-ME 11332, skeleton with lizard in stomach; SMF-ME 11398, skeleton [25,32].
**Locality and Horizon**. Messel Pit, Germany [33]. All known specimens come from the lacustrine "oil-shale" of the Middle Messel Formation (early–middle Eocene, ~48 Ma) [34].
**Emended Diagnosis**. Medium-sized boid snakes, over 2 m in total length, differing from all other snakes in having the following combination of derived features: edentulous premaxilla with bifid vomerine processes; maxilla bearing four labial foramina and 15–18 maxillary teeth; palatine with five teeth and a long maxillary process; 11 pterygoid teeth; dentary with 18–19 teeth; sharp sagittal keel along the basioccipital; the vertebral column with up to 369 vertebrae, of which up to 72 are postcloacal vertebrae.

### 3.2. Brief Anatomical Description

*Eoconstrictor fischeri* was a medium- to large-sized snake (total body length of adult individuals ~200 cm, tail length ~21 cm [25]; Figure 1A), similar to the extant Puerto Rican boa *Chilabothrus inornatus*. The general shape of the skull of *Eoconstrictor* is remarkably similar to that of Neotropical boas, especially *Boa constrictor*.

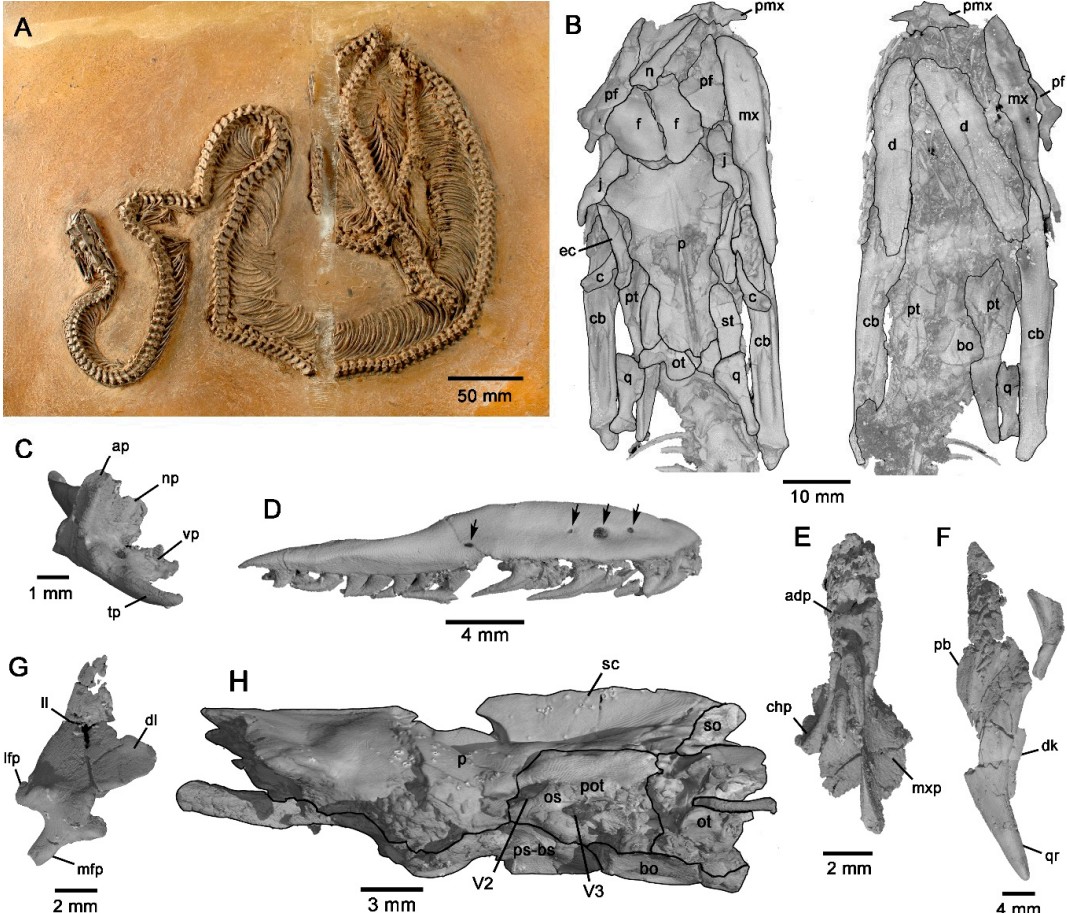

**Figure 1.** Morphology of *Eoconstrictor fischeri*. (**A**), compete skeleton, SMF-ME 11398; (**B**), 3D reconstruction based on CT data of the dorsal (left) and ventral (right) views of the skull of specimen SMF-ME 11398; (**C**), premaxilla in anterolateral view; (**D**), right maxilla in lateral view (arrows indicate labial foramina); (**E**), left palatine in ventral view; (**F**), left pterygoid and ectopterygoid in ventral view; (**G**), left prefrontal in dorsal view; (**H**), braincase in left lateral view. Abbreviations: adp, anterior dentigerous process; ap, ascending process; bo, basioccipital; c, coronoid; cb, compound bone; chp, choanal process; dk, dorsal keel; dl, dorsal lappet; ec, ectopterygoid; f, frontal; j, jugal; ll, lateral lamina; lfp, lateral foot process; mfp, medial foot process; mx, maxilla; mxp, maxillary process; n, nasal; np, nasal process; os, ophidiosphenoid; ot, otooccipital; p, parietal; pb-bs, parabasisphenoid; pf, prefrontal; pmx, premaxilla; pot, prootic; q, quadrate; qr, quadrate ramus; sc, sagittal crest; so, supraoccipital; st, supratemporal; tp, transverse process; V2, foramen for the maxillary branch of the trigeminal nerve; V3, foramen for the mandibular branch of the trigeminal nerve; vp, vomerine process.

The edentulous premaxilla (Figure 1C and Figure S2) bears a well-developed ascending process. The vomerine process is short and the posterior tip is bifid, a unique trait among snakes. The vertical lamina of nasal bone has dorsal and ventral processes (Figure S2), which were in contact with the tip of the ascending process and the nasal process of the premaxilla, respectively. The prefrontal bone exhibits both expanded lateral and dorsal laminae, as in most booids (Figure 1D and Figure S2). It retains only a posterior contact with the dorsal surface of the maxilla through a short, tongue-like lateral foot process. The medial foot process is a remarkably long, finger-like structure, approaching the size observed in boines (Boidae sensu Pyron et al. [30]) and "erycines." Between these processes there is a deep notch for the lachrymal duct, which is open ventrally as in booids. The maxilla bears 18 tooth positions (Figure 1D and Figure S3). It resembles that of *Boa constrictor* in having the anterior maxillary teeth subequal in length to the posterior ones, in contrast to the exceptional long anterior teeth of arboreal boids such as *Corallus* and *Chilabothrus*. Available specimens of *Eoconstrictor* invariably have four labial

foramina of variable size located in the anterior half of the bone (Figure 1D and Figure S3). As in booids, the maxillary process of the palatine arises from the lateral side of the posterior end of the palatine (Figure 1E and Figure S3). The maxillary branch of trigeminal nerve passes dorsally between the palatine and the prefrontal through a groove in the dorsolateral surface of the palatine. The medial edge of the quadrate ramus of the pterygoid crosses dorsally over to the lateral side, forming an oblique keel on the dorsal surface. The posterior tip of the ectopterygoid contacts a well-defined shallow concavity in the lateral surface of the pterygoid (Figure 1F). The frontal exhibits an expanded supraorbital shelf (Figure 1B and Figure S3), as is consistently present in boines, "erycines" and Malagasy boas, thus conferring a square shape to this bone in dorsal view. Dorsally the parietal bears a projecting sagittal crest, which forms an elongate, slender and pointed posterior process that almost totally conceals the sagittal crest of the supraoccipital (Figure 1H and Figure S3). As in most booids, the right posterior opening of the Vidian canal is much larger than the left (Figure S3). The prominent basipterygoid process exhibits an enlarged area for contact with the pterygoid (Figure 1H and Figure S3). Both parabasisphenoid and basioccipital bones have sharp sagittal keels, occupying the posterior third and the entire length of their ventral surfaces, respectively.

### 3.3. Phylogenetic Analysis and Biogeographic Implications

According to both MP (Figure 2) and BI (Figure S4) *Eoconstrictor fischeri* is unambiguously deeply nested in a well-supported, monophyletic Booidea, in congruence with almost all recent phylogenies based on molecular and combined data [9,28,35,36]. Booid synapomorphies include a long medial foot process of the prefrontal, the posterior placement of the maxillary process of the palatine and the large size of the right posterior aperture of the Vidian canal. Among the different clades that comprise Booidea, our analyses posit *Eoconstrictor fischeri* close to Neotropical boas, Boidae (Figure 2 and Figure S4).

The traditional "erycine" group is inferred to be polyphyletic, in line with most recent phylogenies [9,28,35,36]. The recently described small booid *Rageryx schmidi*, also from Messel [11], forms a distinct, well-supported clade together with North American "erycines" *Lichanura* and *Charina*, thus excluding New World "erycines." Interestingly, MP also reveals a close affinity of the other small Messel booids *Messelophis* and *Rieppelophis* to North-Central American ungaliophiine boas.

These phylogenetic results add at first sight a further complication to the booid biogeographic puzzle, since the affinities of *Eoconstrictor fischeri* with Neotropical boas necessarily imply an interchange route between Europe and South America. In agreement with previous estimations based on molecular analyses [8,9,37] our time-calibrated trees indicate that major cladogenetic events for Booidea, including the origin of Neotropical boas, occurred during the Palaeocene–Eocene (Figure 2).

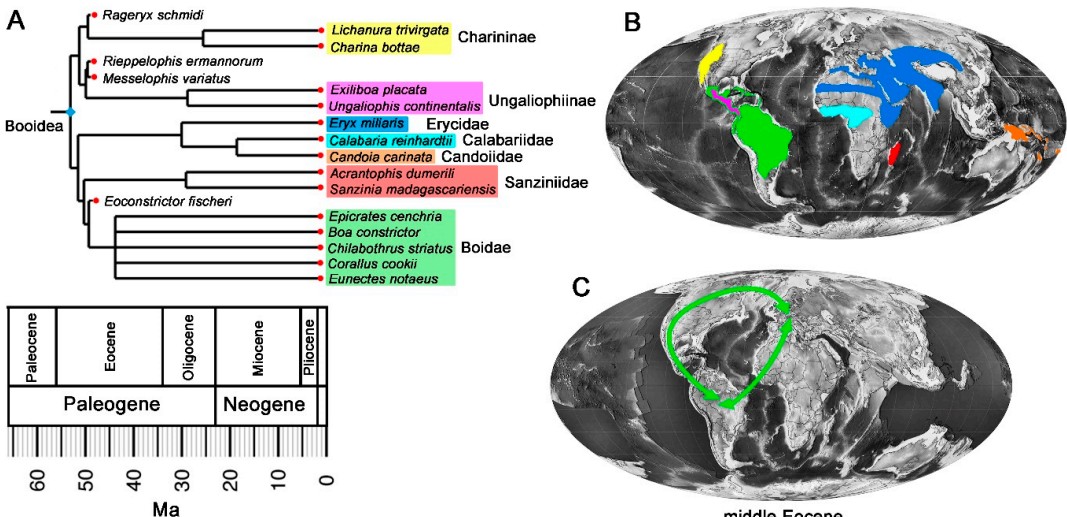

**Figure 2.** Phylogenetic relationships of *Eoconstrictor fischeri* and biogeography of booid snakes. (**A**), Simplified, temporally calibrated tree of booid snakes; (**B**), Current distribution of species of the clade Booidea [10]; (**C**), Palaeogeographic map [38] depicting hypothetical dispersal routes of boine snakes during the middle Eocene. See also electronic Supplementary Mterials, Figure S4 for further details.

### 3.4. Infrared Reception in Messel Booids

Booids and pythons display several pits with different arrangements in rostral and labial scales [4,23]. Boas in particular display only labial pits, which are located at the caudal margin of the labial scales, more precisely in the interstitial skin between contiguous scales (Figure 3A). Heat receptors located in the fundus of pit organs are innervated by different branches of the trigeminal nerve depending of their location, and are profusely irrigated by a capillary network supplied by arteries ultimately derived from the internal carotid (Figure 3B). These nerves and blood vessels pierce the jaw bones (maxilla, compound bone and dentary), enabling their cross-sectional area to be studied. Although a complete picture of the heat reception and pit organs anatomy in snakes is far from being achieved, we know that jaw foramina are clearly enlarged in species of boas and pythons with pits organs compared to species without them [22]. Since neurovascular structures have morphogenetic primacy, we assume that the size of the jaw foramina is proportional to the cross-sectional area of neurovascular tissues that supply the receptors in pit organs.

In our training data set we found a highly significant correlation between the size of maxillary labial foramina and the incidence of pit organs in the upper jaw ($P = 0.00352$), and between the size of the lower jaw foramina and the incidence of pit organs in the lower jaw ($P = 0.0107$) (Figure 3D). Applying the logistic model to the size of the foramina in *Eoconstrictor*, we calculate a high probability that *Eoconstrictor* had pit organs in the upper jaw ($P = 0.922$) but not in the lower jaw ($P = 0.016$). Conversely, our model provides no support for the presence of pit organs in the coeval, small-sized booid species of Messel (*Rieppelophis*: $P = 0.13$ and $0.15$ for upper and lower pit organs, respectively; *Messelophis*: $P = 0.01$ and $0.28$ for upper and lower pit organs, respectively; *Rageryx*: $P = 0.005$ and $0.06$ for upper and lower pit organs, respectively).

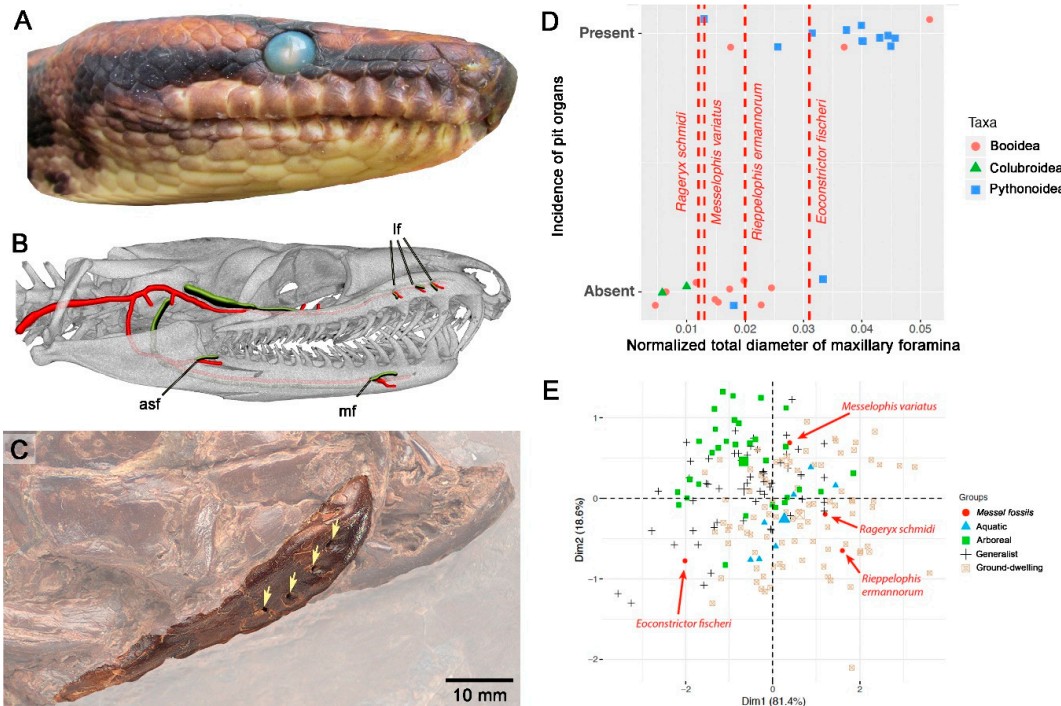

**Figure 3.** Infrared pit organs in *Eoconstrictor fischeri* and its habitat preferences. (**A**), head of the rainbow boa *Epicrates cenchria* anterolateral view showing the pit organs located in supralabials and infralabial scales; (**B**), 3D reconstruction based on CT data of the skull of *Epicrates cenchria*, with the nervous (green) and blood (red) main supplies for pit organs emerging from the foramina located in maxilla and lower jaw; (**C**), dorsolateral view of the right maxilla of *Eoconstrictor fischeri* (SMF-ME 1002) showing four labial foramina (yellow arrows); (**D**), relationship between the incidence of pit organs in the upper jaw and the normalised total diameter of maxillary foramina (see text for details). Fossil taxa from Messel shown as vertical lines for the appropriate foramen size; (**E**), graph of the two principle components, with ecological classes and fossil taxa distinguished. See also Supplementary Mterials for further details. Abbreviations: asf, anterior surangular foramen; lf, labial foramina; Lt, *Lichanura trivirgata*; mf, mental foramen.

### 3.5. Macrohabitat Preferences in Messel Booids

Neotropical boas inhabit a wide range of habitats including forests of various kinds, shrublands and savannas. Accordingly, these snakes display diverse macrohabitat preferences including generalist (*Boa* spp., *Epicrates* spp.), aquatic (*Eunectes* spp.) and arboreal forms (*Corallus* spp., *Chilabothrus* spp.) [39,40]. Previous studies based on ecomorphological traits of extant boids have suggested that ancestral forms of this clade were stout, medium to large-sized snakes with a short tail and occupied semi-arboreal to arboreal macrohabitats [40]. Arboreal boids are small to medium size forms and exhibit laterally compressed light bodies [40–42], all features that can be inferred from osteology.

According to our multivariate statistical analyses, probabilities of group membership of *Eoconstrictor fischeri* are: generalist (61.7%), ground-dwelling (18.1%), arboreal (14.9%) and aquatic (5.2%) (Figure 3E). In other words, the probability that *Eoconstrictor* spent considerable time on the ground (was not arboreal or aquatic) is around 79.8%. The generalised anatomy observed in the skeleton of *Eoconstrictor* is in agreement with this result. The small Messel booids *Rieppelophis ermannorum* and *Rageryx schmidi* are inferred to be ground-dwelling forms (83% and 91%, respectively), and these taxa also show skeletal traits consistent with this inference [11,36]. *Messelophis variatus*, in contrast, plots in a part of PC-space distinct from the others (positive PC2 score). It is intermediate in size between *Eoconstrictor* and the other two. It also has a much longer tail (at least 26% of SVL in HLMD-Me 15013, where some probably small part of the tail is also missing), although not as long as in many

arboreal species, where relative tail length exceeds 30% [24]. Its body is long due to the high number of vertebrae (c. 300 trunk vertebrae) but threadlike. LDA produces no single compelling hypothesis as to its macrohabitat preference: arboreal (29%), generalist (38%) or ground-dwelling (31%). As relative tail length rises, so will the probability that *Messelophis* was arboreal in habits.

## 4. Discussion

### 4.1. Phylogeny and Biogeographic History

The quantitative analysis of biogeographic history has advanced tremendously in recent years, e.g., [43]. However, these methods are yet of little use in studying the history of booids, because Messel is, in a sense, the only booid snake assemblage. To be sure, a large number of fossil booid snake taxa has been described from the Palaeogene of Europe [44,45] and North America [46] and to a lesser extent Africa [47] and South America [48]. However, almost all of these taxa are based exclusively on isolated (frequently mid-trunk) vertebrae, and their phylogenetic affinities are therefore virtually unconstrained. To exemplify the problems of phylogenetic interpretation, Smith [49] studied associated cranial elements and extensive sampling of the entire vertebral column in two late Eocene species from North America, previously considered to pertain potentially to the same genus of 'erycine' booid, e.g., [50]. He showed that these species not only are not 'erycines', but they are not even closely related to one another. One, *Calamagras weigeli*, is apparently related to the dwarf boa clade Ungaliophiinae, whereas the other, *Ogmophis compactus*, is apparently related to the Mexican Burrowing Python, *Loxocemus bicolor* [11,49]. Since several booid lineages (total clades of Ungaliophiinae, Charininae, Boidae) have their oldest, or near-oldest (if *Titanoboa* is a stem boid [51,52]), records in Messel, this leads to the appearance that they originated in Europe and dispersed to the New World (or beyond), rather than the other way around. Given the total absence of evidence from North America, and Africa however, this cannot be accepted at face value.

Several of the Messel booid lineages are estimated to have diverged from extant snakes near the Palaeocene–Eocene boundary, coincident with the prolonged period of global warming and hyperthermals around the Palaeocene–Eocene boundary [53]. Range expansion, as inferred for a number of lizard taxa in North America [38], could have promoted diversification [54], especially if accompanied by colonisation of Europe. Regardless, we consider it most probable that Booidea originated in the New World, where the centre of species diversity still lies, and dispersed to Europe, producing the lineages at Messel. Testing that hypothesis will require the discovery of well-preserved early Palaeogene fossils from the New World. The locality of Fossil Lake [55] as well as rare, associated material from other sites [56,57] indicate that this is possible.

Assuming the total clade of Boidae itself has a South American origin, it remains to be established by what route *Eoconstrictor* arrived in Europe. Taking into account the long-term isolation of South America from the Upper Cretaceous to the Neogene, two alternative dispersal scenarios can explain our results. A South America-to-Europe dispersal route through Africa, which necessarily entails a transatlantic dispersal, was postulated by various researchers from the Late Cretaceous to the Palaeogene [58–60]. The other possibility is a South America-to-Europe route via North America, which is supported by compelling evidence about the faunal dispersal route between North America and Europe during the Palaeogene [61–64]. The lack of fossils from Africa and North America with known phylogenetic relations does not allow us to discriminate between these possibilities at present.

### 4.2. Labial Pits in Extant Snakes

Labial pits are one of the most distinctive features of booid and pythonid snakes, for these organs, together with the facial pits of crotaline vipers, make them capable of perceiving infrared radiation, uniquely among vertebrates. The photons coming from the environment of an animal are a mix of reflected photons, typically in the ultraviolet and visible spectrum, and photons emitted as

blackbody radiation, typically in the infrared [65]. Organs for infrared reception therefore give access to a completely new visual field representing the thermal environment.

The circumoral scales of all examined booids and pythons exhibit specialised receptors called terminal nerve masses, or TNMs [3,4]. Each is the expanded, pyramidal terminus, with abundant mitochondria, of the larger branch of the axon of a pseudobipolar neuron whose soma is located in either the ophthalmic or the maxillomandibular ganglion of the trigeminal nerve [3]. The other branch of the axon of this neuron projects to a specialized part of the myelencephalon called the lateral descending tract and nucleus of the trigeminal nerve, or LTTD [66]. From there, signals are passed via relays to the optic tectum of the contralateral side (similar to visual signals from the lateral eyes), where they map spatiotopically with signals from the lateral eyes onto the tectal surface [3]. It is therefore believed that visible light and infrared radiation are integrated into a single 'broadband' [3] image of the environment.

The receptors are exceedingly sensitive, with a rise in temperature of 0.003 °C or less capable of producing a signal (modulating the background firing of the neurons) [3,23]. The rich capillary beds of the pit organs are thought to help cool the TNMs rapidly and avoid 'afterimages' [3]. TNMs have been documented, and may occur in a concentrated fashion, in the circumoral epithelium of booid species that do not exhibit labial pits, such as *Boa constrictor* and *Eunectes murinus* [3,67]. Indeed, their occurrence is surely responsible for the ability of booids lacking pits, such as the aforementioned species and *Lichanura trivirgata*, to perceive radiant energy [23]. Crucially, however, in pit organs the nerve supply is greater, the receptors more abundant, the capillary network denser, and the epidermis thinner than in surrounding areas [23,68]. This is the basis for the correlation we found above between the size of jaw foramina (which carry the branches of the trigeminal nerve as well as the blood supply) and the incidence of pits.

Because the radiant heat receptors and the LTTD are unique to snakes capable of perceiving radiant energy and are present even in species of Booidea lacking pit organs, it is likely that this system was minimally present in the common ancestor of Booidea (and for similar reasons that of Pythonidae). Whether this system is present also in more basally branching taxa such as *Xenopeltis* and *Loxocemus*, e.g., [35], much less other alethinophidian snakes, has yet to be examined. While the ability to sense radiant energy may by itself be advantageous, pits offer further advantages. First, the much greater density of receptors in the fundus (base) of the pit confers greater sensitivity [23]. Second, because the orifice is always narrower than the fundus, it becomes possible to perceive also directionality and movement [3]. Yet the distribution of labial pits in Booidea, especially their absence in *Boa* and *Eunectes*, together with the great variability in the number, location and shape of these pits, has suggested that they may have arisen multiple times even in this clade, e.g., [3,23,65].

*4.3. Eoconstrictor and the Evolution of Labial Pits*

Fossil evidence bearing on the problem has until now been wanting. The inferred presence of labial pits in *Eoconstrictor fischeri* therefore gives new insight into their pattern of evolution. First, it shows that a species close to the ancestor of crown Boidae possessed labial pits, making it possible that their absence in extant taxa like *Boa* and *Eunectes* represents loss. This would turn the evolutionary question on its head. Second, it shows that the first documented labial pits are located in the upper jaw, rather than in the lower jaw or both simultaneously. Finally, it shows that labial pits evolved very early (in a temporal sense) in the history of Booidea, so that they may have played a larger role in the diversification of the group than hitherto suspected. Until now the timing of their origin has been little constrained.

It is considered that pit organs may confer a selective advantage for different reasons, which may differ depending on the habitat, among other factors. Better visual discrimination of prey has featured most prominently in functional studies [3,65]. Clearly, for predators on homeothermic prey (such as mammals or birds) this may be especially important, particularly so if the predator is nocturnal, as may be inferred for *Eoconstrictor* given the analyses of Hsiang et al. [35]. At the same time, it has been

demonstrated experimentally that visible light, as opposed to infrared, modalities may dominate in directing prey strikes, and it is likely that both modalities are often used simultaneously [65]. Other potential selective advantages have received less attention. These include predator avoidance [65,69], thermal microhabitat discrimination [3,4,23,70], and even the selection of ambush sites [71]. In the latter case, it was considered that the relatively cool background of arboreal perches may assist in the discrimination of flying, homeothermic prey. More generally, labial pits may confer advantages 'in the general life of snakes . . . as enhancers of the visual senses of their possessors' ([3] p. 293).

As the earliest booid snake in which pit organs have been documented, *Eoconstrictor fischeri* illuminates the context in which they arose. The use of pit organs in the detection of homeothermic prey would be a potential function in *Eoconstrictor*, but available dietary data are inconsistent with that assumption. The large specimen described by Greene [72], which in fact appears to be *Eoconstrictor*, has a crocodylian, probably *Diplocynodon* sp. based on size, in its stomach. (Coils of vertebrae cover the head and tail, so that distinguishing characteristics of the two species [73] cannot be studied. Furthermore, the plate on which it is conserved is impregnated with fibreglass, so that even high-resolution X-radiographs yielded no insight.) A juvenile *Eoconstrictor* had consumed a basilisk lizard, *Geiseltaliellus maarius* [32]. A specimen of a small mammalian carnivore [74] and a bird [75] were suggested to have been regurgitated by a large constrictor, but in light of the recognition of a greater diversity of constrictors at Messel, it is unclear to which species these specimens should be attributed. Thus, available direct evidence suggests that poikilotherms were important in the diet of *Eoconstrictor*, despite the availability of abundant homeothermic species of appropriate size, such as lipotyphlan mammals [76] and flightless birds [77]. Given the extensive behavioural adaptations to maintain a constant activity temperature, it is not out of the question that pit organs are also useful in targeting other poikilothermic amniotes as well. However, *Eoconstrictor* does not support the hypothesis that the earliest pit organs were exclusively used to catch homeothermic prey.

The detection and avoidance or deterrence of homeothermic predators is also not supported. Messel is unusual in that large, homeothermic predators are absent from the assemblage [33,78]. While this absence might partly reflect a taphonomic filter, it should be noted that large herbivores, especially basal perissodactyls, are abundant [79]. Furthermore, Mayr [77,80] summarised a rich assemblage of flightless birds at Messel, a fact he attributed to an original absence of large terrestrial predators there. Thus, there is no evidence that the labial pits of *Eoconstrictor* played a role in the detection or deterrence of homeothermic predators.

Finally, the use of pit organs in arboreal ambush sites is theoretically possible. Flying, homeothermic vertebrates, especially bats, were abundant at Palaeolake Messel [81]. However, our analysis of habitat preferences suggests a terrestrial way of life, not stenotopically arboreal. Thus, there is no reason to believe that the upper pit organs of *Eoconstrictor* were useful in finding such sites (if this were possible [65]) or catching prey at them. In sum, there is no evidence that the pit organs of *Eoconstrictor* played a role in predator–prey relations.

As emphasised by Krochmal et al. [70], Goris et al. [3] and others, the ability to sense radiant energy may play many other, less spectacular roles in the life of a snake, and *Eoconstrictor* suggests that it is amongst this panoply of possibilities that the functional origin of pit organs within Booidea is to be sought, like, perhaps, the origin of infrared detection itself. At the same time, the advantages noted above that are conferred by pit organs in comparison with mere infrared receptors—the ability to perceive directionality and movement—highlight a conundrum. If *Eoconstrictor* did not specialise on homeothermic prey and had no need to avoid large homeothermic predators, then pit organs of the modern type would seem overbuilt. Thus, the limits of our conclusions with regard to the pit organs of *Eoconstrictor* should be emphasised. The high density of infrared receptors (TNMs) and vascularisation suggested by our results, which today are uniquely found in pit organs, say nothing about the morphology of those organs. In particular, the soft tissue surrounding the inferred concentrations of receptors is unconstrained, and we do not know the form of the orifice (aperture). In consequence, the extent to which *Eoconstrictor* could discriminate directionality and movement is

unknown. Finally, we must emphasise again that the number of specimens in which gut contents are preserved is low.

Amongst extant booids (as well as pythons), it is only medium to large-sized species that bear conspicuous pit organs, and they all occupy terrestrial habitats and frequently consume large endothermic prey such as mammals and birds. As our results showed that small booid species from Messel lacked pit organs, they support the existence of a common pattern since the earliest evolutionary history of this clade: pits only occur in larger species. If so, there may exist a noteworthy correlation between size, habitat use and diet that influenced (and still influences) the evolution of pit organs in booid snakes.

Further questions about the origin of the pit organs remain unanswered, such as the importance of their distribution in the circumoral area. *Eoconstrictor* apparently only had pit organs in the upper jaws, as in extant *Morelia viridis*, whereas other extant species, such as *Antaresia childreni*, only have them in the lower jaws. What different roles the exact distribution, not to mention the shape and number, of pit organs might serve in boas and pythons remain unknown.

Although the ecomorphology of *Eoconstrictor* could be taken as ancestral for Boidae, caution is yet warranted, given the scant knowledge about other fossil boids. Indeed, if boid affinities and piscivorous feeding ecology of the giant aquatic snake *Titanoboa cerrejonensis* from the Palaeocene of Colombia [51,52] are confirmed, the ecomorphology and habitat preferences of early boas must have been more diverse than previously thought.

## 5. Conclusions

The Messel snake assemblage can be seen as the only Palaeogene snake assemblage in the sense that only in Messel can the phylogenetic relations of the component species be studied in detail. As such, it adds significantly to the morphological diversity and palaeobiology of the earliest booids. Our phylogenetic results reinforce the diversity of booid lineages that inhabited the vicinity of Palaeolake Messel [82], and the extant relatives of the Messel taxa are noteworthy for being found exclusively in the New World. Messel preserves a diverse snake fauna in the early stages of its evolution, with different ecomorphs occupying different macrohabitats. The presence of pit organs in *Eoconstrictor* furthermore complements other information on diet in this species (summarised in [32]) and suggests that neither predator–prey relations nor the use of arboreal ambush sites were prominent at the origin of these unique sensory organs. Rather, the origin may lie in the broader life of the species.

**Supplementary Materials:** The following are available online at http://www.mdpi.com/1424-2818/12/3/100/s1, Document S1 (Supplementary figures S1–S4, details of the phylogenetic analyses and specimens examined), Phylogenetic matrix (NEXUS format), Data S1 (pit organ survey table), Data S2 (habitat survey table), and R scripts for statistical analyses.

**Author Contributions:** A.S. and K.T.S. designed the study, collected the data, conducted the phylogenetic and statistical analyses, and wrote the manuscript. All authors have read and agreed to the published version of the manuscript.

**Funding:** This research was funded by the Deutscher Akademischer Austauschdienst (DAAD) and the Ermann Foundation.

**Acknowledgments:** The authors thank Stephan Schaal (Senckenberg), Klaus Winkelmann (Fossilien- und Heimatmuseu, Messel), and Claudia Koch (Alexander Koenig Museum, Bonn) kindly provided access to collections and/or CT facilities. Anika Vogel and Sabine Köster (Senckenberg) generally assisted during A.S.'s extended visit, and Sabine Köster conducted some of the digital segmentation of SMF-ME 11398. Renate Rabenstein (Senckenberg) kindly conducted high-resolution X-radiography of the specimen of *Eoconstrictor* in the collection of the Fossilien- und Heimatmuseum, Messel. The paper also benefited from reviews by anonymous reviewers.

**Conflicts of Interest:** The authors declare no conflict of interest.

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
