# Peer review of "Exquisitely Preserved Fossil Snakes of Messel: Insight into the Evolution, Biogeography, Habitat Preferences and Sensory Ecology of Early Boas"

_diversity, doi:10.3390/d12030100_

Round 1

Reviewer 1 Report

The Messel locality provided unusually well-preserved early-middle Eocene boas (Booidea) whose skeletons are preserved in anatomical position. Authors resolved a debate concerning the taxonomical position of „Palaeopythonfischeri. This species differs from the type species of Palaeopython (P. cadurcensis) as well as Paleryx (P. rhombifer) and therefore, they proposed a new combination Eoconstrictor fischeri (Schaasl, 2004). The detailed study of cranium revealed the presence of labial pit organs in the upper jaw and authors discuss the earliest evidence of incorporation of infrared spectrum into visual systems of snakes. The combined molecular and morphological data show that Eoconstrictor was  closely related to Neotropical boas and discuss a South America-to-Europe dispersal event. All these results are novel and important for understanding of Booidea evolution.

Methods are appropriate, author use modern imaging techniques (micro-CT). The phylogenetical analysis is based on both morphological, mainly osteological, characters in combination with DNA sequences taken from GenBank. Standard, widely used software (TNT, Mr. Bayes, R) was used for reconstruction of the maximum parsimony and divergence times  estimation. All results are logically arranged and clearly presented. Systematic Palaeontology is fine, description brief but sufficient. However, there are several issues that should be improved in figures to be consistent with the main text:

Figure 1:

1) The Figure 1B with 3D reconstruction of Eoconstrictor skull from dorsal and ventral aspects should be larger. Although bones outlines are clearly visible, CT-scans seem too tiny to me. The same for a braincase from the left lateral view (Figure 1G).

2) There is no information on prefrontal in figure caption of Figure 1. Prefrontal is placed as D (but the same for maxilla).

3) There is no reference on Figure 1A in the text.

4) Several abbreviations are not explained in Figure 1 caption: pb, sc, P, Os, Pot, Ps-Bs, So, Ot, Bo, V2, V3. They are explained in Figure S3 but I would prefer to supplement explanations in every figure caption. Authors use capital letters in abbreviations of bones - Pot, Ot, Ps, Bs. In Supplementary file 3 are used small letters – pot, ot, ps… Please, unify.

Fugure 2: Authors should supplement A, B, and C to be in agreement with the figure caption of the Figure 2. In time scale below the tree a unit (Ma) should be supplemented.

Figure 3: In figure caption, there is explained „Lt, Lichanura trivirgata“ but I do not see it in Figure 3.

page 3:

„We calculated the common logarithm of all measurements and conducted principle components analysis (PCA) on the logged data (figure S9).“ There is no Figure S9 in the text or in Supplementary files.

„… because they is not present in…“; write „are“ instead of „is“

page 5:

„The vertical lamina of nasal bone has dorsal and ventral processes (Figures 1b and S2)…“ It can not be seen in Figure 1B. This is visible only in S2-H.

page 6:

„Dorsally the parietal bears a projecting sagittal crest, which forms an elongate, slender and pointed posterior process that almost totally conceals the sagittal crest of the supraoccipital (Figures 1h and S3).“ Authors refer to the Figure 1h but there is no Figure 1H. They should refer to the (Figure 1B and S3-Q,S,T).

„The prominent basipterygoid process exhibits an enlarged area for contact with the pterygoid (Figures 1h and S3).“ Authors should refer to the (Figure 1B and S3-P,R,T,U).

Table S1 and S2: Units (mm, cm) are absent in tables. 

Reviewer 2 Report

The paper of Scanferla and Smith provides important information on the systematics, anatomy, and biogeography of Messel booids. The taxonomic and biogeographic implications are sound and reasonable. The quality of Figures is nice and it will certainly augment our current knowledge of European fossil boas.

I have only minor comments to suggest:

- The authors mention sometimes boines but it is stated that they follow Pyron et al. (2014). Please clarify if these “boines” are the equivalent of “Boidae sensu Pyron et al. (2014)”.

- Page 9. Discussion: “To be sure, a large number of fossil booid snake taxa has been described from the Palaeogene of Europe [e.g., 43]”.

Here i would suggest instead some more relevant references, such as Szyndlar and Rage 2003, Szyndlar et al. 2008:

Szyndlar, Z., Rage, J.-C., 2003. Non-erycine Booidea from the Oligocene and Miocene of Europe. Institute of Systematics and Evolution of Animals, Polish Academy of Sciences, Kraków, 111 pp.

Szyndlar, Z., Smith, R. and Rage, J.-C. 2008. A new dwarf boa (Serpentes, Booidea, ‘Tropidophiidae’) from the Early Oligocene of Belgium: a case of the isolation of Western European snake faunas. Zoological Journal of the Linnean Society 152:393–406.

- Etymology: Eos was in fact in Greek mythology, the goddess that brought the dawn to Earth every morning. You may also want to mention the original spelling “Έως”.

- In Fig. 2, “middle Eocene” should have not the first letter of “middle” capitalized. Again, in the same Fig., there is a mistake in the timeline and subdivision of epochs: Neogene consists of Miocene and Pliocene, so it should terminate there; after that is the Quaternary.

- The authors should use "Palaeo" instead of "Paleo" as it is a European journal. So far both types of spellings are present in the text - please be consistent.

- Second paragraph of Introduction, the binomen Eoconstrictor fischeri is not italicized. This is also the case in other places of the text. Please modify.

- between ages, it is more appropriate to use large dash “–” and not a small one “-“. The authors use both formats throughout the text. Please be consistent.

- In the references, Georgalis and Scheyer is not in press  but published already: 2019, 112:383–417.

Also, in the same reference, the genus name Palaeopython should be italicized.

Besides these minor remarks, I consider this as an important contribution and I am highly recommending it for publication after some minor revision.
